# A synonymous KCNH2 polymorphism and methadone trough level influence QTc prolongation in Kelantanese Malay recipients of methadone maintenance therapy (MMT) in Malaysia

**Muhammad Irfan Abdul Jalal**[1]*, **Muhammad-Redha Abdullah-Zawawi**[1], **Nurfadhlina Musa**[2], **Basyirah Ghazali**[3], **Zalina Zahari**[4], **Nasir Mohamad**[5]

1 UKM Medical Molecular Biology Institute (UMBI), Cheras, Kuala Lumpur, Malaysia, 2 Human Genome Centre, School of Medical Sciences, Universiti Sains Malaysia, Kubang Kerian, Malaysia, 3 Institute for Research in Molecular Medicine (INFORMM), Universiti Sains Malaysia, Kubang Kerian, Kelantan, Malaysia, 4 Faculty of Pharmacy, Universiti Sultan Zainal Abidin, Besut Campus, Besut, Terengganu, Malaysia, 5 Faculty of Medicine, Sultan Zainal Abidin University, Medical Campus, Kuala Terengganu, Terengganu, Malaysia

* irfan.abduljalal@ukm.edu.my

## Abstract

Potassium voltage-gated channel subfamily H member 2 (*KCNH2*) polymorphisms have been found to influence the heart-rate adjusted QT (QTc) intervals. We investigated the association between *KCNH2* polymorphisms and QTc intervals among Malay opioid-dependent methadone maintenance treatment (MMT) recipients. A cross-sectional study was conducted involving 111 patients with stable methadone dosage for at least 6 months attending several methadone clinics in Kelantan, Malaysia between March 2011 and October 2012. Those with cardiac structural defects, recipients of other QTc-prolonging pharmacotherapeutic agents, had aggressive behavior or other active psychiatric illnesses, chronic medical and surgical ailments and who were unable to communicate in Malay and English were excluded. The Fridericia-corrected QTc intervals were recorded using a 12-lead electrocardiogram. DNA samples were extracted from peripheral blood leukocytes and genotyped using nested allele-specific polymerase chain reaction for these four *KCNH2* polymorphisms: 1539C>T, 1956T>C, 2350C>T, and 2690A>C. Mean QTc interval is 408 ms (SD: 24). Molecular docking was performed on all four *KCNH2* polymorphisms to investigate the impact of the nucleotide changes on methadone binding. Based on multiple regression analysis, only 1539T>C polymorphism ($\beta_{adjusted}$: 10.506 (95% CI:0.846, 20.166), p=0.033; recessive model), serum methadone trough ($\beta_{adjusted}$: 0.025 (95% CI: 0.006, 0.043), p=0.009), potassium ($\beta_{adjusted}$: -8.756 (95% CI: -15.938, -1.575), p=0.017) and magnesium ($\beta_{adjusted}$: -106.226 (95% CI: -159.291, -53.161), p<0.001) levels were significantly associated with mean QTc. Molecular docking

**Data availability statement:** The full dataset for this study has been shared and made publicly available on the Harvard Dataverse repository (doi: https://doi.org/10.7910/DVN/WMK7HL).

**Funding:** This research work was supported by the USM 'Research University Grant (RUI)' (Grant ID: 1001/PPSP/812056).

**Competing interests:** The authors have declared that no competing interests exist.

analysis resulted in good binding-energy values between the 1539C > T and methadone, with the formation of hydrophobic and π–π stacking interactions, suggesting that 1539C > T was the newly discovered SNP involved in QTc prolongation. In conclusion, the 1539C > T *KCNH2* polymorphism is associated with QTc prolongation in our MMT recipients, necessitating QTc monitoring to prevent methadone-associated cardiotoxicity in this Malay MMT population.

## Introduction

Methadone is widely used in opioid dependence treatment and was introduced in Malaysia in 2005 to curb HIV transmission [1]. This initiative has shown positive outcomes, as HIV incidence in Malaysia declined from 28.4 per 100,000 population in 2002 to 11.7 per 100,000 in 2014 [2]. However, methadone is linked to serious adverse effects, including QTc prolongation and torsade de pointes (TdP) [3,4]. Factors such as high methadone doses, stimulant use [5], liver dysfunction, CYP3A4 inhibitors, and electrolyte imbalances contribute to QTc variability, though unidentified factors may also be involved [6].

Pharmacogenomics has identified genetic variations influencing drug response, including polymorphisms in ion channel genes linked to long QT syndromes (LQTS) and TdP [7–9]. One such gene, the potassium voltage-gated channel subfamily H member 2 (*KCNH2*) gene, encodes the human ether-à-go-go related gene (hERG) potassium channel, which regulates cardiac repolarization. Polymorphisms in *KCNH2*, such as 1039C > T (Pro347Ser) and 2350C > T (Arg784Trp), are associated with drug-induced TdP [10–13]. Methadone and other hERG-blocking drugs may pose higher LQTS risk in individuals carrying these variants [14,15].

*KCNH2* polymorphisms in opioid-dependent individuals remain underexplored in Southeast Asia, with most studies focusing on general populations or small cohorts [13,16]. Research on their interaction with methadone is especially limited, including data on their frequency in Malay populations [17]. Identifying these variants could improve methadone safety by enabling personalized dosing strategies [16].

This study hypothesizes that *KCNH2* variations contribute to methadone-related QTc prolongation. Understanding these genetic risk factors could help tailor MMT dosing, enhancing patient safety and treatment efficacy [18]. To date, only Hajj et al. have investigated *KCNH2* polymorphisms in MMT recipients, linking the 2690A > C polymorphism and methadone dosage to QTc prolongation in a French cohort [16]. However, no study has examined this association in the Kelantanese Malay population, making this the first endeavour to explore the relationship between *KCNH2* polymorphisms and QTc intervals in opioid-dependent Kelantanese Malay MMT recipients.

## Methods

### Study design and participants

This cross-sectional study involved 111 Malay opioid-dependent patients who are free from cardiac structural defects and attended the Hospital Universiti Sains

Malaysia (HUSM) Psychiatric Clinic and four other MMT clinics (SAHABAT-Kota Bharu, Wakaf Bharu, Ketereh and Selising), Kelantan, Malaysia between 1st of March 2011 and 31st of October 2012. The reason for such racial selectivity is to minimize cross-ethnic variations in the *KCNH2* polymorphism frequency. The study enrolled participants who were taking prescribed similar daily doses of methadone as described in previous research [19].

The study's inclusion criteria are (a) patients aged 18 years and above; (b) had stable methadone dosage for at least six months during the MMT; (c) of Kelantanese Malay ancestry of up to three generations; (d) a history of good compliance to the Directly Observed Therapy (DOT)/ takeaway program; and (e) consented to study participation. We excluded the patients if they had cardiac structural defects (e.g., atrial or ventricular septal defects (ASD/VSD), valvular heart diseases, aortic coarctation), received other QTc-prolonging pharmacotherapeutic agents (e.g., haloperidol, amitriptyline, amiodarone, nelfinavir, efavirenz), had aggressive behavior or other active psychiatric illnesses, chronic medical and surgical ailments and patients who were not English or Malay proficient. The ethics approval for the research protocol was granted by the Universiti Sains Malaysia (USM)'s Human Research Ethics Committee (HREC), Kelantan, Malaysia (Reference number: USMKK/PPP/ Ethics.Com/2010(19), approval date: 01/02/2010). This study was conducted in accordance with the ethical principles outlined in the Declaration of Helsinki for research involving human participants and all participants provided written informed consent prior to their inclusion in the study.

Initially, 140 eligible MMT patients regularly attended the recruitment centres and were, therefore, available for study participation. However, 29 patients were excluded for the following reasons: participation refusal (n = 10), mixed ancestry (n = 9), incomplete follow-up data (n = 6), and more than 20% missing clinical and genotyping data (n = 4). The participants were recruited using convenience sampling due to the paucity of suitable patients available for study recruitment. The flow of participant recruitment is summarized as a Strengthening the Reporting of Observational studies in Epidemiology (STROBE) [20] diagram in Fig 1.

This study is registered with the ClinicalTrials.gov registry (ID: NCT03603158), and the full dataset is available from the Harvard Dataverse repository (doi: https://doi.org/10.7910/DVN/WMK7HL). The Strengthening the Reporting of Genetic Association Studies (STREGA) guideline was followed to improve the quality and clarity of reporting for this study [21].

## Sample size calculation

The sample size for detecting the prevalence of hERG polymorphisms was calculated using a single-proportion formula [22]:

$$n = (z / \Delta)^2 \, p \, (1 - p) \tag{1}$$

Where p is the proportion of subjects with the SNPs, z = 1.96, and $\Delta$ is the precision. Based on the information obtained from Koo et al., the largest sample size required is for detecting 1539T > C SNP [17]. The proportion of Malay subjects with C allele for 1539T > C SNP is 0.441, and $\Delta$ is fixed at 0.095. Hence, the biggest sample size required is therefore 105 subjects. Sample size calculation for detecting the differences in mean QTc between different genotypes in all *KCNH2* polymorphisms under study could not be performed due to the absence of satisfactory information from previous literature to aid such an endeavor.

## Clinical data collection

The consented patients were interviewed to gather socio-demographic information, history of drug addiction, drug dependency patterns, other drug usage, psychiatric illnesses, and relevant treatment-related issues. Subsequently, a validated Malay version of the Subjective Opioid Withdrawal Scale (SOWS) questionnaire [23] was administered to assess opioid withdrawal symptoms.

Five ml of venous blood was collected from each participant to ascertain serum biochemical profiles ($K^+$, $Mg^{2+}$, $Ca^{2+}$), methadone trough levels, and *KCNH2* genotyping. Urine dipstick (FastScreenTM Drug Combo Test 6-in-1. Reszon

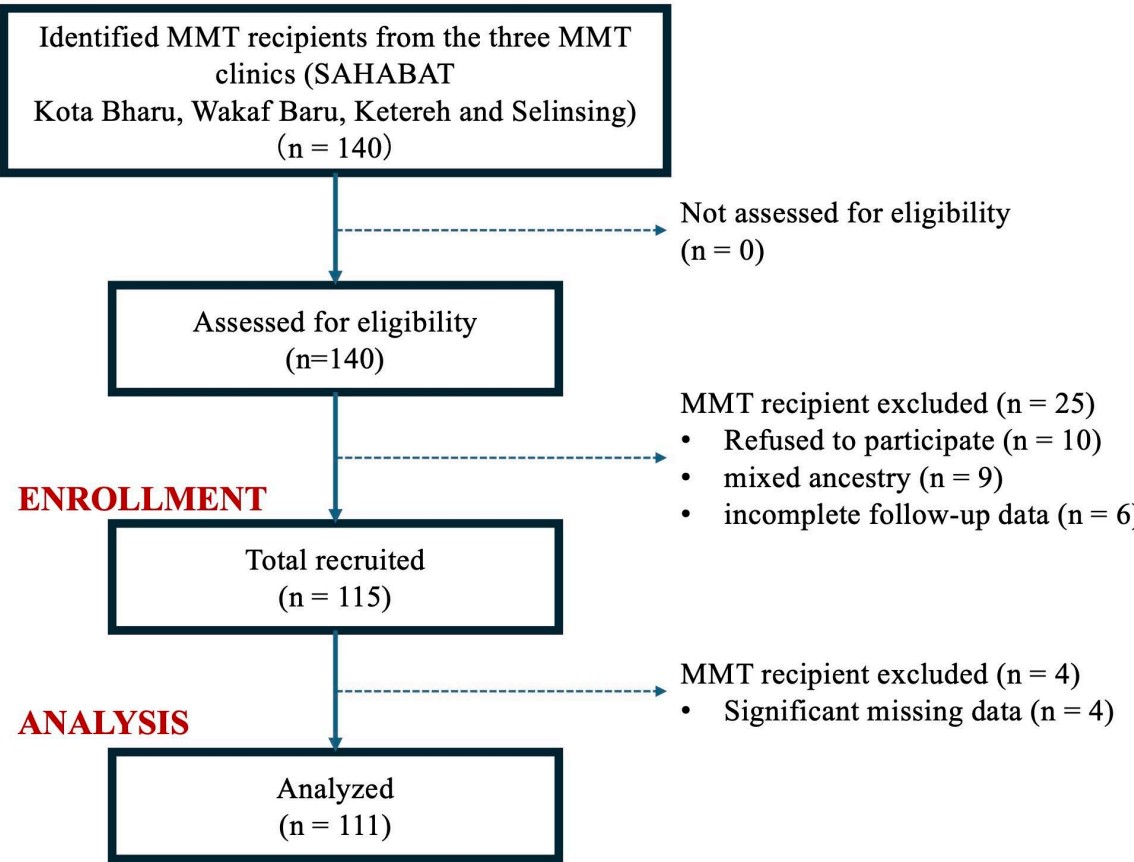

**Fig 1. The STROBE flow diagram summarizing the progress of each phase of this research.**

Diagnostics, Subang Jaya, Malaysia) tests were performed for methamphetamine, cannabis or marijuana, methylenedioxyl methamphetamine (MDMA) and benzodiazepines. Patients were required to provide samples at the urine collection point, accompanied by the clinic staff. The color and temperature of the specimen were recorded.

A standard 12-lead electrocardiogram (ECG) was performed by an experienced internist (NM) using validated Welch Allyn CP 50™ ECG (Welch Allyn Australia Pty Ltd., New South Wales, Australia) machines. All ECGs were recorded by the same instrument and printed at a paper speed of 25 mm/s and voltage of 10 mm/mV. QT interval was quantified in milliseconds (ms) utilizing the method elucidated in Wedam et al. [24] and Ehret et al. [6]. Fridericia's formula ($QTc = QT/(RR)^{0.33}$) was then employed to calculate the QTc interval [25]. No patients reported any prior history of TdP, syncope, or seizure. All personnel involved in $QT_c$ measurement were masked to information on other biochemical parameters and serum methadone trough concentrations.

### Serum methadone trough level measurement

The quantification of serum methadone trough level (methadone $C_{trough}$) was executed based on the previously described protocol [26]. Serum methadone trough concentration Methadone $C_{trough}$ was measured using a validated Methadone ELISA kit invented by the researchers of Pharmacogenetic and Novel Therapeutic Cluster, INFORMM that has optimal sensitivity [27]. The quantification limit of this method is 50 ng/ml and it can be thus typically employed in a routine clinical context.

## DNA extraction

DNA extraction and *KCNH2* genotyping were conducted at the Pharmacogenetic Research Laboratory, Institute for Research in Molecular Medicine (INFORMM), Kelantan, Malaysia. About 2.5 ml blood samples for genotyping were collected in sodium citrate tubes from all participants after QTc measurement, and these were rapidly transferred to -20 °C freezer in ice boxes. Genomic DNA from peripheral leucocytes was extracted from 200 µl of each participant's whole blood using the QIAamp DNA Blood Mini Kit (Qiagen Gmbh, Hilden, Germany) according to the manufacturer's instructions. The quantity and quality of the extracted genomic DNA were evaluated by using the NanoDrop ND-1000 Spectrophotometer (NanoDrop Technologies, Inc. Wilmington, USA) with measurements done at 260 and 280 nm, and these were stored in -20 °C freezer.

## Design of allele-specific PCR for *KCNH2* polymorphism detection

All samples were subsequently genotyped for the following *KCNH2* polymorphisms; rs1805120 (1539C > T, F513F), rs1137617 (1956T > C, Y652Y), rs12720441 (2350C > T, R784W), and rs1805123 (2690A > C, K897T) using a two-step nested allele-specific polymerase chain reaction (PCR). The reason why 1539C > T, 1956T > C, and 2690A > C were chosen is that they are the most commonly encountered *KCNH2* SNPs in the general Malay population, and 2690A > C is directly associated with QTc prolongation in methadone recipients [16,17]. For 2350C > T, several studies verified its presence and association with drug-associated QT prolongation and TdP [13,28].

In the first PCR (PCR1), we amplified exons 6, 8, 9 and 11 of *KCNH2*. We subsequently utilized the PCR1 products to identify the presence of 4 SNPs using allele-specific primers in the second PCR (PCR2) performed in 2 parallel sets/reactions to ensure optimal product size; set A (2690A>C and 1956T>C) and B (2350C>T and 1539C>T). For PCR2, the primers were meticulously designed to possess a mismatch at specific 3′-ends and were specific to both the variant and wild-type DNA sequence at a particular locus. The primers were also manipulated to distinguish single nucleotide alteration/allele at a particular locus during PCR amplification. We utilized the Basic Local Alignment Search Tool (BLAST) tool, which is available at http://www.ncbi.nlm.nih.gov/blast, to verify the primers' specificities [29]. All primers were synthesized by Invitrogen© USA, and purchased from a local vendor, and the melting temperature ($T_m$) was approximated as follows: $T_m = 2(A+T) + 3(C+G)$, and the optimal $T_m$ was identified via gradient PCR method employing the Veriti 96-well Thermal Cycler® (Applied Biosystem, Foster City, CA). S1 Table provides further information on the primers used in PCR1 and PCR2 reactions.

## PCR protocols

The final reaction mixture for PCR1 consists of 2 µl of DNA, 10.5 µl of primer mix (volume and concentrations of each primer were shown in S1 Table plus 2.5 µL of $dH_2O$), and 12.5 µl of ready-made commercial MyTaq™ Mix (Bioline®, London, UK) mastermix, a total volume of 25 µL per reaction. The PCR amplifications of exons 6, 8, 9 and 11 were performed using GeneAmp® PCR system 9700 Perkin Elmer (Applied Biosystems, Foster City, CA) under the following thermocycle profiles; an initial pre-denaturation phase for 95 °C for 1 minute and this is succeeded by 25 cycles of denaturation at 95 °C for 1 minute, annealing at 65 °C for 15 seconds and extension 72 °C for 10 seconds. For the last step, the final extension phase at 72 °C for 7 minutes was also incorporated.

The PCR1 products were separated using ethidium bromide-stained 2% agarose gel (Promega Corporation, Madison, USA) electrophoresis in mini electrophoresis tanks (Primo Submarine Gel system™. ThermoQuest, Holbrook, New York, USA) at 130V for 90 minutes. The PCR products were subsequently visualized under ultraviolet (UV) light and photographed. The laboratory personnel who conducted the experiments were blinded to QTc measurements and other biochemical profiles of study participants.

Two µl of PRC1 amplicons, which were diluted to 1:20, served as PCR2 DNA templates. The total volume of the master mix for each set A and B reaction is 10 µl and this consists of $dH_2O$ (set A: 6.0 µl; set B:5.75 µl), 1x reaction buffer (2.5 µl;

both sets), MgCl$_2$ (set A: 0.5 µl; set B: 0.75 µl) and dNTP (0.5 µl; both sets) and *Taq* polymerase (0.5 µl; both sets). For the primer mix, the total volume is 13 µl for both sets A and B, which comprises wild-type and mutant primers (S1 Table) and dH$_2$O (9 µl). The final volume for each parallel PCR reaction is 25 µL.

The thermocycler profile for PCR2 is similar to PCR1 except for the following: annealing at 69 °C for 30 seconds and extension at 72 °C for 4 seconds. The PCR2 products were separated and visualized using the same modality as PCR1 (S1 Table).

### Direct DNA sequencing of *KCNH2* polymorphism

The PCR1 products were submitted for direct sequencing. QIAquick PCR purification kit (Qiagen, USA) was used to purify PCR products. Sequencing was performed using 3130XL genetic analyzer DNA sequencer (ABI, USA). The published sequences for *KCNH2* in the NCBI [accession number: NC_000007.13] were subsequently used to compare the DNA sequencing results.

### Data and statistical analysis

The frequencies of the *KCNH2* allele and genotype were calculated using the gene counting technique based on the samples' observed genotype numbers. The Hardy–Weinberg equilibrium (HWE) departure was statistically tested by comparing the observed and expected frequency of homozygous genotypes using the Levene and Haldane's exact test based on the sum equally likely or more extreme samples (SELOME) [30–32]. An R package, HardyWeinberg version 1.59, implemented on the R platform (R Core Team, 2017. Vienna, Austria) was deployed to achieve this. The magnitude of HWE deviation was calculated and represented using Lindley's α [33], and if the value was more than 0.5, the SNP was definitely dropped from further analyses. The formula for Lindley's α is:

$$\alpha = 0.5 \ \log(4P_{aa} * P_{AA} / (P_{Aa})^2); \ -\infty < \alpha < +\infty \tag{2}$$

where $P_{aa}$, $P_{AA,}$ and $P_{Aa}$ represent the proportion of each genotype [33,34].

Simple linear regression analysis was used to screen for variables (*KCNH2* polymorphisms, serum methadone trough levels, serum potassium, magnesium, calcium levels, patient's age and gender) that may be potentially associated with QTc intervals. The independent predictors with p values below 0.25 in the univariable analysis were selected for multi-variable model building using backward and forward stepwise multiple linear regression ($p_{entry} = 0.05$, $p_{removal} = 0.10$). The presence of multicollinearity and effect modifiers were also respectively examined by using variance inflation factor (VIF)'s threshold of less than 10 and assessing the significance of the 2-way interaction term for *KCNH2* genotypes and serum methadone trough level. Model assumptions (normality, homoscedasticity, and independence of residuals) were assessed using Studentized residuals versus predicted value plots and a Q-Q plot of residuals. The IBM Statistical Package for the Social Sciences (SPSS) Version 29 (Released September 2022. Armonk, NY: IBM Corp) was utilized to carry out statistical analyses. All estimates were reported with their 95% confidence intervals (Cis) and significance threshold were fixed at 0.05 (two-tailed).

### Identification of functional domains in KCNH2 protein sequences

Proteins comprise various domains, each associated with distinct functions. Therefore, examining SNPs in the KCNH2 at the domain level may offer valuable insights into molecular processes that may affect QTc prolongation among MMT recipients. Before constructing the protein structure model, an integrated database of protein families, domains, and functional sites called InterPro was utilized to determine the domain architecture of KCNH2 protein sequences [35]. The location of the mutations was subsequently mapped across the functional domains and interdomains of the KCNH2 protein using Illustrator for Biological Sequences version 2.0 [36].

## Protein structure modeling of KCNH2 polymorphisms

To analyze the impact of mutations on the protein structure, protein structure modeling was carried out on the KCNH2 protein sequence for identified SNPs, such as 1539C>T, F513F (synonymous); 1956T>C, Y652Y (synonymous); 2350C>T, R784W (missense); and 2690A>C, K897T (missense). Firstly, the protein sequence of KCNH2 was obtained from the UniProt database [37]. The sequences of KCNH2 proteins were then mutated by manually altering amino acids of both missense variants R784W and K897T from arginine (R) and lysine (K) to tryptophan (W) and threonine (T), respectively. However, the alteration in the coding sequences *KCNH2* (1539C>T and 1956T>C) did not result in any change to the amino acid. Therefore, the KCNH2 F513F and Y652Y proteins were regarded as wild-type (WT) proteins since their structures remained unaltered. To model the protein structures, a Support Vector Machine (SVM) was employed in conjunction with the SWISS-MODEL server [38]. The SVM algorithm integrates interface conservation, structural clustering, and various template features to generate a Quaternary Structure Quality Estimate (QSQE) and a Global Model Quality Estimate (GMQE) [38]. The accuracy of the structural assessment was measured through GMQE and QSQE scores, which both range between 0 and 1. After the assessment, SWISS-MODEL was employed to conduct a Ramachandran plot analysis on the models to predict the stereochemical quality of the predicted protein structures. Finally, the protein structures were visualized and analyzed using UCSF Chimera software [39].

## Mutational effect prediction of missense variant 2350C>T (R784W) and 2690A>C (K897T)

Missense variants can result in either benign or deleterious effects, as they may influence protein stability and potentially disrupt the binding between the proteins and ligands (drugs). Therefore, by employing statistical and ML-based servers like Predicting the Effects of Mutations on Protein Stability (PremPS) [40], SVM-based approach DUET [41], and mutation Cutoff Scanning Matrix (mCSM) [41], we measured the mutational effects of the missense variants 2350C>T (R784W) and 2690A>C (K897T) on protein stability, which may support their impact on methadone binding compared to 1539C>T (F513F) and 1956T>C (Y652Y).

## Molecular docking analysis

To associate SNPs with QTc prolongation among MMT recipients, the binding affinity between methadone and KCNH2 proteins with 2350C>T (R784W), 2690A>C (K897T), 1539C>T (F513F) and 1956T>C (Y652Y) was assessed by performing protein-ligand docking analysis using CB-DOCK2 [42,43]. CB-DOCK2 is a docking server that employs CurPocket, an algorithm based on protein surface curvature, to identify cavities and facilitate molecular docking using AutoDock Vina [43]. Before docking, ligand preparation involved obtaining the structure of methadone (CID: 4095), dextromethadone (CID: 643985), and levomethadone (CID: 22267) from PubChem [44], followed by converting the structure from SDF to the MOL2 file format using OpenBabel [45]. Following protein modeling, the CASTp Server was deployed to predict the active sites/cavities in the KCNH2 protein [46]. To assess the impact of *KCNH2* polymorphisms on drug binding, the predicted structures of the MT and WT KCNH2 proteins were docked with methadone, dextromethadone, and levomethadone using CB-DOCK2, with contact residues for docking specified to the predicted active sites/cavities. All three drug structures were docked within the cavity using the same docking size and region to compare their binding affinities to the pocket of the MT and WT KCNH2 proteins. The binding energy (kcal/mol) of the MT and WT KCNH2 complexes with methadone, dextromethadone, and levomethadone was subsequently evaluated. The binding affinity of WT KCNH2 served as the control, while the binding affinity of MT KCNH2 was compared to that of WT KCNH2 to determine the impact of the polymorphism on drug-protein binding stability.

## Results

### Patients' clinical characteristics

A total of 111 subjects were included in this study. The average age of the MMT recipients was 36.02 (SD: 7.02, range: 20–59) years old. Nearly all study participants were males (male: n = 109 (98.2%), female: n = 2 (1.8%)). Consequently, the gender-stratified analysis could not be conducted due to an inadequate number of female participants.

Based on urinary drug screening, 62 (55.9%) subjects were drug-free, and in those with positive drug screen, the most frequently abused drug was morphine (n = 5, 4.5%), although 7 (6.3%) subjects were multiple drug abusers, 36 (32%) subjects were injectable drug users and 1 (1.1%) subject with unknown drug screening status. No other significant co-morbidities that may affect QTc prolongation (e.g., hypertension, thyroid dysfunction, renal and hepatic dysfunction) were observed in the participants. The summary of relevant clinic-demographic characteristics and SOWS scores is given in Table 1.

### *KCNH2* genotype and allele frequencies

The genotyping of DNA samples from the 111 drug users was successfully accomplished for the four *KCNH2* SNPs. The genotype frequencies observed in this study were similar to those predicted by the HWE for 1539C > T ($p_{exact(SELOME)}$ = 0.4513) and 2690A > C ($p_{exact(SELOME)}$ = 0.1101) only, whilst the observed genotype frequencies for 1956T > C (minor allele frequency (MAF): 0.730) and 2350C > T (MAF: 0.036) significantly deviated from the expected genotype frequencies under HWE ($p_{exact(SELOME)}$ = < 0.001 (Lindley's α = 1.077) and 0.004 (Lindley's α = 0.860), respectively. Full results are available upon request). Consequently, both 1956T > C and 2350C > T were excluded from further statistical analyses. Allele frequencies for the remaining *KCNH2* polymorphisms and comparisons with results from prior studies are shown in Table 2. Examples of amplified SNPs are available in Fig 2.

### *KCNH2* polymorphisms and QTc intervals

Based on the genetic additive model (Table 3), there are significant differences between 1539C > T polymorphisms and QTc. However, only CT and TT genotypes have significant QTc differences based on post-hoc multiple comparison whilst the other comparisons were insignificant. No significant differences in QTc were found among 2690A > C genotypes. Due to inconclusive findings, the analyses were modified, and genetic recessive and dominant models were used instead to further examine the associations between the two SNPs and QTc intervals.

Using simple linear regression analysis, only 1539C > T SNP, methadone trough levels, serum potassium, and magnesium concentrations reached statistical significance. These covariates were subsequently selected for multivariable model building using multiple linear regression. The effects of 2690A > C SNP and other variables on QTc are not statistically significant and, therefore, were not considered. Table 4 indicates that all 4 variables retained their significance and, therefore, were included in the predictive model for mean QTc. No significant multiplicative interaction effects or multicollinearity were found between the covariates.

Model assumption check revealed that all multiple linear regression assumptions were fulfilled, and no interaction terms were deemed significant for inclusion in the final predictive model. The predictive model for mean QTc is thus given by the regression equation below:

$$\text{Mean QT}_c = 529.592 + 10.506 \, (\text{TT genotype} = 1, \, \text{CC or CT} = 0) + 0.025 \, (\text{Methadone trough level})$$
$$- 8.756 \, (\text{Potassium level}) - 106.226 \, (\text{Magnesium level})$$

(3)

We can thus interpret that a subject with TT genotype experiences 10.506 ms longer mean QTc compared to TT or CT genotype after adjustment was made for the effects of serum methadone trough concentration and serum potassium and magnesium levels. This QTc prolongation exceeds the 5–10 ms threshold that regulatory agencies, such as the US Food And Drug Agency (FDA) and International Council for Harmonisation (ICH), consider for potentially concerning QT prolongation risk in drug safety studies [47]. Although this threshold is typically applied to drug-induced effects, even small QTc increases may be clinically relevant, particularly in individuals with additional genetic risk factors for arrhythmias.

**Table 1. Summary of clinico-demographic and pharmacogenetic characteristics of study subjects (n = 111).**

| Variables | n (%) | Mean (SD) | Range |
|---|---|---|---|
| Age (years) | | 36.02 (7.02) | 20 - 59 |
| Gender | | | |
| Male | 109 (98.2) | | |
| Female | 2 (1.8) | | |
| Duration of drug use (years) | | 6.3 (2.60) | 0.66 - 13.00 |
| Daily methadone dose (mg/day) | | 34.01 | 20 - 190 |
| Serum methadone trough (ng/ml) | | 258.84 (222.62) | 10.96 - 1404.38 |
| $QT_c$ intervals (milliseconds) | | 408 (24) | 369 - 500 |
| Sodium (mmol/L) | | 137.9 (3.2) | 131 - 147 |
| Potassium (mmol/L) | | 4.4 (0.6) | 3.1 - 6.6 |
| Magnesium (mmol/L) | | 0.83 (0.08) | 0.61–1.00 |
| Calcium (mmol/L) | | 2.25 (0.13) | 2.00–2.52 |
| SOWS scores | | 22.94 (7.2) | 15 - 55 |
| rs1805120 (1539C>T, F513F) | | | |
| 1539CC | 60 (54.1) | | |
| 1539CT | 25 (22.5) | | |
| 1539TT | 26 (23.4) | | |
| rs1137617 (1956T>C, Y652Y) | | | |
| 1956TT | 18 (16.2) | | |
| 1956TC | 24 (21.6) | | |
| 1956CC | 69 (62.2) | | |
| rs12720441(2350C>T, R784W) | | | |
| 2350CC | 105 (94.6) | | |
| 2350CT | 4 (3.6) | | |
| 2350TT | 2 (1.8) | | |
| rs1805123 (2690A>C, K897T) | | | |
| 2690AA | 90 (81.1) | | |
| 2690AC | 18 (16.2) | | |
| 2690CC | 3 (2.7) | | |

## Protein structure models of WT and MT KCNH2

Protein structure models of WT KCNH2 (1539C>T (F513F) and 1956T>C (Y652Y)) demonstrated 96.37% sequence identity with potassium voltage-gated channel subfamily H member 2 (O35219.1.A) structure generated by AlphaFold v2, whereas both MT KCNH2 (2350C>T (R784W) and 2690A>C (K897T)) indicated 96.29% identity to the similar template. The homology models exhibited an exceptionally conserved structure with a high degree of sequence identity (>96%) and high structural similarity, as indicated by the root mean square deviation (RMSD) values of 0.001 Å after superimposing WT and MT KCNH2 to the template, respectively. Hence, it is suggested that the predicted benign missense SNPs, 2350C>T (R784W) and 2690A>C (K897T), do not cause any conformational changes in the KCNH2 protein. Structure assessment revealed 76.40% of all residues in Ramachandran favored regions, with 4.52% being rotamer outliers and 12.53% Ramachandran outliers. Although the favored regions were less than 90%, the models were reliable enough to perform virtual screening, as the models showed conserved features, especially in functional domain regions (Fig 3).

The KCNH2 protein comprised the Per-Arnt-Sim (PAS) domain (37–132 aa), ion transport domain (409–668 aa), cyclic nucleotide-binding domain (742–860 aa), and cytoplasmic domain (664–1159 aa). Both synonymous SNPs, 1539C>T

**Table 2. Observed frequencies of *KCNH2* alleles and their 95% confidence interval (n = 111).**

| Polymorphic alleles | Location | Observed allele frequencies (95% CI)[+] | Minor allele frequencies from prior studies (Races; Sample size, [References]) |
|---|---|---|---|
| **rs1805120 (1539C > T, F513F)** | Exon 6 | | |
| *1539C* | | 0.495 (0.429, 0.561) | **Minor allele: T** |
| *1539T* | | 0.505 (0.439, 0.571) | 0.441 (Singaporean Malays;114, (Koo, Ho, and Lee 2006))[a] |
| | | | 0.351 (Singaporean Chinese;265 (Koo, Ho, and Lee 2006))[a] |
| | | | 0.327 (Singaporean Indians;139, (Koo, Ho, and Lee 2006))[a] |
| | | | 0.258 (Han Chinese;297, (Wang et al. 2009))[a] |
| | | | 0.280 (Japanese; 50 (Iwasa et al. 2000))[a] |
| | | | 0.270 (Caucasian; 32, (Paulussen et al. 2004))[a] |
| | | | 0.820 (Danish;46, (Larsen et al. 2001))[a] |
| **rs1805123 (2690A > C, K897T)** | Exon 11 | | **Minor allele: C** |
| | | | 0.119 (Singaporean Malays;114, (Koo, Ho, and Lee 2006))[a] |
| *2690A* | | 0.892 (0.851, 0.933) | 0.047 (Singaporean Chinese;265 (Koo, Ho, and Lee 2006))[a] |
| *2690C* | | 0.108 (0.067, 0.149) | 0.158 (Singaporean Indians;139, (Koo, Ho, and Lee 2006))[a] |
| | | | 0.02 (Japanese;50 (Iwasa et al. 2000))[a] |
| | | | 0.140 (American; 92, (Yang et al. 2002))[a] |
| | | | 0.230 (Caucasian; 32, (Paulussen et al. 2004))[a] |
| | | | 0.160 (Finnish;201, (Laitinen et al. 2000))[a] |
| | | | 0.723 (Turkish;74, (Atalar et al. 2010])[a] |
| | | | 0.235 (German;1030, (Bezzina et al. 2003))[a] |
| | | | 0.04 (African American; 100, (Anson et al. 2004))[a] |
| | | | 0.341 (French;82, (Hajj et al. 2014))[b] |

+ For this study; [a]In Healthy population; [b]In methadone recipients.

(F513F) and 1956T > C (Y652Y), were located within the ion transport domain region. Additionally, a substitution from arginine to tryptophan at position 784 was observed within the cyclic nucleotide-binding domain, and a substitution from lysine to threonine occurred at position 897 within the cytoplasmic domain region.

### Effects of missense mutations on KCNH2 protein stability

Previously, the lack of differences between favorable residues of WT and MT KCNH2 indicates that the core structure of the aforementioned domains was not affected by the 2350C > T (R784W) and 2690A > C (K897T), and none of these variants could cause conformational changes that would affect the functions of KCNH2. However, both 2350C > T (R784W) and 2690A > C (K897T) have been demonstrated to impact the stability of the KCNH2 protein. The missense SNPs were highly predicted to destabilize the surface of the KCNH2 protein, as indicated by the predicted stability change (ΔΔG) values (Table 5). Thus, despite both missense SNPs being benign, they appeared to destabilize the protein structure of KCNH2, which could potentially disrupt the contact residue of drug binding in MMT recipients with these SNPs, leading to shorter QTc intervals.

### Drug binding affinity evaluation between WT and MT KCNH2 with methadone

To infer the possible role of *KCNH2* polymorphisms in affecting QTc prolongation among MMT recipients, the binding affinity of methadone, levomethadone, and dextromethadone to each KCNH2 protein cavity was predicted via a molecular docking study. The structural analysis of KCNH2 identified 14 cavities, with Phe513 (F513F), Tyr652 (Y652Y), and both Arg784 (R784) and Trp784 (W784) being determined as binding pocket residues, except for Lys897 and Thr897 (K897T). Among these cavities, the target cavities of 1539C > T (F513F) (13, 50, 52, and 75) and 1956T > C (Y652Y) (5, 15, 40, and 76) were predicted to be situated within the ion transport domain, while 2350C > T (R784W) was found in cavities 1, 12, 20, 68, 69, and 90, which are located within the cyclic nucleotide-binding domain (Table 6).

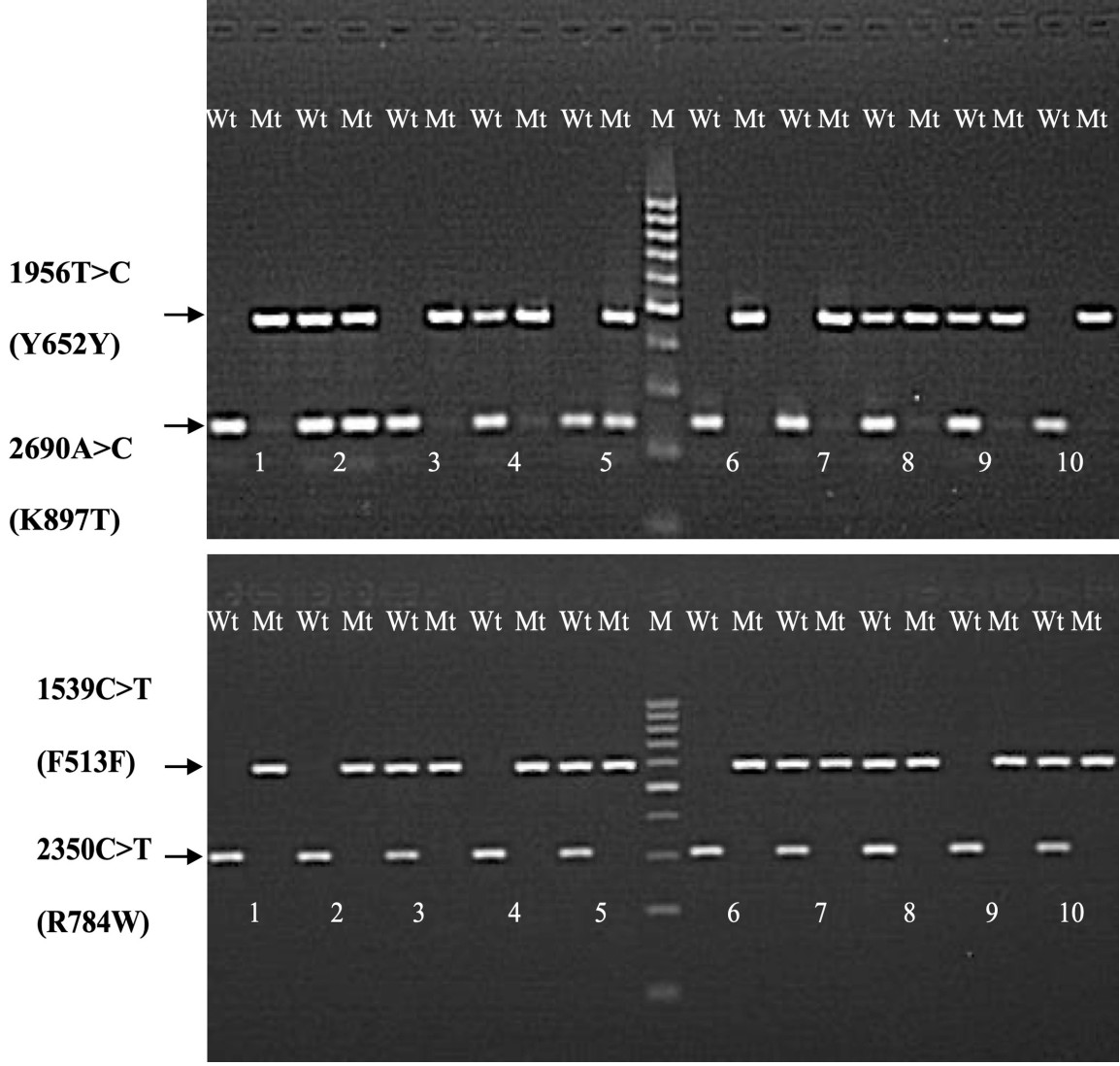

**Fig 2. The PCR products for the four KCNH SNPs using representative samples obtained from ten MMT recipients.** M: DNA ladder; Wt: Wild type; Mt: Mutant.

Methadone was docked and redocked with levomethadone and dextromethadone into a similar cavity or region of interest using a similar docking center and size. K897T was excluded from molecular docking analysis as the mutation was not determined to be located at the binding pocket of KCNH2. All three types of methadone exhibited more negative binding energy values to 1539C>T (F513F) compared to 1956T>C (Y652Y), 2350C>T (R784), and 2350C>T (W784). Overall, the drug binding energy score of F513F ranges from 6.5 to 6.9 kcal/mol, followed by Y652Y (5.8–6.1 kcal/mol), and R784 (3.7–3.8 kcal/mol). Out of the four mutations, W784 was also shown to have the highest binding energy score, suggesting a destabilizing effect on the KCNH2 protein, leading to ineffective drug binding (Table 5). In this study, we observed the interaction between Phe513, Tyr652, Arg784, and Trp784 with methadone as a ligand to decipher the possible role of polymorphisms in affecting the binding affinity of the drugs (Fig 4).

Interestingly, the racemic methadone mixture, levomethadone, and dextromethadone were all found to interact with Phe513, forming three hydrophobic interactions and a π–π stacking in all predicted cavities (Fig 5). The strongest

**Table 3. Comparisons of mean QTc between different genotypes based on the additive genetic models (n = 111).**

| Variables | Mean QTc (SD) (in ms) | F statistics $(df_1, df_2)$ | P values | Post-hoc comparisons (QTc differences; p value (Schaeffe) |
|---|---|---|---|---|
| **rs1805120 (1539C > T, F513F)** | | 4.072 (2, 108) | **0.020** | |
| CC | 410.68 (23.02) | | | TT – CT: 15.62 (95% CI: 1.74, 29.50); **0.023** |
| CT | 402.80 (23.69) | | | TT – CC: 7.74 (95% CI: -8.82, 24.30); 0.512 |
| TT | 418.42 (24.85) | | | CC – CT: 7.88 (95% CI: -6.19, 21.95); 0.384 |
| **rs1805123 (2690A > C, K897T)** | | 4.040 (2)[c] | 0.133 | – |
| AA | 404 (40)[a] | | | |
| AC | 415 (28)[a] | | | |
| CC | 393[a] (-)[b] | | | |

[a]Median (IQR); [b]Cannot be calculated due to low frequency of CC genotype; [c]Kruskall-Wallis H statistics (degree of freedom); $df_1$: Numerator degree of freedom; $df_2$: Denominator degree of freedom.

binding affinity is observed in cavity 13, with binding energies of -6.9 kcal/mol, -6.9 kcal/mol, and -6.8 kcal/mol, respectively.

On the other hand, Tyr652 only forms hydrophobic interactions with all three methadone compounds (racemic mixture, levomethadone and dextromethadone), with energy scores lower than the Phe513-methadone binding. Meanwhile, both Arg784 (WT KCNH2) and Trp784 (MT KCNH2) had the lowest binding energy score with a range between -3.7 to -4.8 kcal/mol by forming a hydrophobic bond with all methadone within cavity 90 (Fig 6). Based on our structural and docking study, it is strongly suggested that the 1539C > T (F513F) polymorphism is associated with QTc prolongation in MMT recipients. However, more in-depth validation is required to confirm this association in future studies.

## Discussion

This research primarily aims to evaluate the link between *KCNH2* genetic variants and QTc intervals in opioid-dependent MMT recipients who received stable daily methadone dosages. Of the four genetic polymorphisms we studied, one was identified frequently in our Malay MMT population (1539C > T (F513F), and one was less frequent (2690A > C (K897T)). No definitive conclusion could be about the prevalence of 1956T > C (Y652Y) and 2350C > T (R784W) minor alleles since both SNPs violated the HWE principle, which can be attributed to sampling bias secondary to genotyping error, population stratification, or population inbreeding [48]. This is supported by the high positive values of Lindley's α (exceeding 0.5) for both 1956T > C and 2350C > T SNPs, signifying excess proportions of homozygotes [33,34].

Besides, this is also the first report confirming the association between rs1805120 (1539C > T, F513F) SNP and QTc interval prolongation in methadone recipients of the Kelantanese Malay descent, a subethnic group of Malay population living in the North-eastern region of Peninsular Malaysia that has genetic similarities to the Semang (one of the Malaysian aboriginal ethnic) and Indian populations [49], based upon the recessive model. We believe this finding has not been documented by any previous study conducted in other populations. We also established the first predictive model for QTc prolongation in Malay MMT recipients that also incorporates other significant clinical covariates such as methadone trough level, serum potassium and magnesium levels. We also found no significant multiplicative interaction effects among significant QTc predictors (1539C > T, methadone trough level, serum potassium, and magnesium concentrations), thus further refining the findings of Hajj and colleagues [16]. In contrast, Zerdazi and colleagues demonstrated significant multiplicative association between the intronic variant, rs11911509 in *KCNE1* encoding the beta subunit of the potassium channel, and methadone dosage and QT$_c$ prolongation in opioid-dependent French subjects on MMT [50].

Furthermore, based on the recessive genetic model, we also established that MMT patients with CC genotype (2690A > C SNP) have more reduced mean QTc intervals than MMT recipients having AA/AC genotypes (406.333 ms vs

**Table 4. The associations between HERG SNPs (under recessive and dominant models), clinico-demographic factors, and QT$_c$ interval (n = 111).**

| Predictors | Simple linear regression | | | Multiple linear regression[c] | | |
|---|---|---|---|---|---|---|
| | Regression Coefficients (95% CI) | t statistics | p values | Regression Coefficients (95% CI) | t statistics | p values |
| **Age (years)** | 0.013 (-0.016, 0.043) | 0.885 | 0.378 | | | |
| **Gender (Female)** | -4.321 (-39.073, 30.450) | -0.246 | 0.806 | | | |
| **rs1805120 (1539C > T, F513F)** | | | | | | |
| **[RECESSIVE MODEL]** | – | | | | | |
| CC+CT | 13.305 (2.681, 23.930) | 2.482 | **0.015[b]** | 10.506[d] (0.846, 20.166) | 2.156 | **0.033** |
| TT | | | | | | |
| **rs1805120 (1539C > T, F513F)** | | | | | | |
| **[DOMINANT MODEL]** | – | | | | | |
| CT+TT | 3.157 (-7.899, 14.213) | 0.566 | 0.573 | | | |
| CC | | | | | | |
| **rs1805123 (2690A > C, K897T)[a]** | | | | | | |
| **[RECESSIVE MODEL]** | – | | | | | |
| AA+AC | -1.954 (-30.473, 26.566) | -0.136 | 0.892 | | | |
| CC | | | | | | |
| **rs1805123 (2690A > C, K897T)** | | | | | | |
| **[DOMINANT MODEL]** | – | | | | | |
| CC+AC | -8.756 (-20.447, 2.936) | -1.484 | 0.141 | | | |
| AA | | | | | | |
| **Methadone trough level (ng/mL)** | 0.021 (0.001, 0.041) | 2.030 | **0.045[b]** | 0.025 (0.006, 0.043) | 2.679 | **0.009** |
| **Potassium (mmol/L)** | -13.483 (-21.047, -5.919) | -3.533 | **0.001[b]** | -8.756 (-15.938, -1.575) | -2.417 | **0.017** |
| **Magnesium (mmol/ L)** | -117.297 (-173.035, -61.559) | -4.171 | **<0.001[b]** | -106.226 (-159.291, -53.161) | -3.969 | **<0.001** |
| **Calcium (mmol/ L)** | 11.780 (-24.521, 48.082) | 0.643 | 0.571 | | | |

[a]AA+AC vs CC: 406.333 ms vs 408.287 ms; [b] Included in multiple linear regression analysis; [c] $R^2$ = 0.276, Constant term = 529.592 (95% CI: 476.116, 575.067; p value <0.001), [d]Adjusted mean of QT$_c$ CC+CT vs TT: 405.773 ms vs 416.279 ms. Evaluated at the following covariate values: $Mg^{2+}$ (0.827), $K^+$ (4.377), Methadone trough (258.842).

408.287 ms), albeit being statistically insignificant (p = 0.892). Nevertheless, our results still parallel the findings of Hajj et al. [16], who demonstrated that based on the recessive model (AA+AC vs AC), the mean QTc for the CC group was much shorter than the AA+AC group (383 ms vs 419 ms, p = 0.004). This is further corroborated by Gouas and co-workers [51], who demonstrated that the C allele of 2690A > C polymorphism is more frequently encountered in French healthy subjects with short QTc intervals. Besides, these observations were strengthened by Bezzina et al. [52] and Marjamaa et al. [53], who verified such associations. Our insignificant finding is attributed to the small study sample size, which is evident from the low proportion of the minor C alleles in our MMT subjects (MAF: 0.108 vs 0.341, Table 1), resulting in the reduction of our study power. Furthermore, this discrepancy can also be explained by the different formulas used for calculating QTc (Bazett's formula) by Hajj et al. [16]. Vandenberk and colleagues [25] demonstrated that Fredericia's method is superior to Bazett's formula in QTc computation since the former had enhanced capacity for predicting 30-day and 1-year all-cause fatality. Hence, the predictive potential of 2690A > C on QTc prolongation remains an open debate for our MMT population.

The mechanisms explaining the basis of our observed association between 1539C > T (F513F) SNP and QTc interval are, however, still unclear. We speculate that a synonymous SNP may alter co-translational protein folding, which results in protein structural modification and substrate specificity [54]. Furthermore, synonymous *KCNH2* nucleotide alterations that decrease GC content, scarce codon, and the sum of mRNA secondary structure result in mRNAs that are less

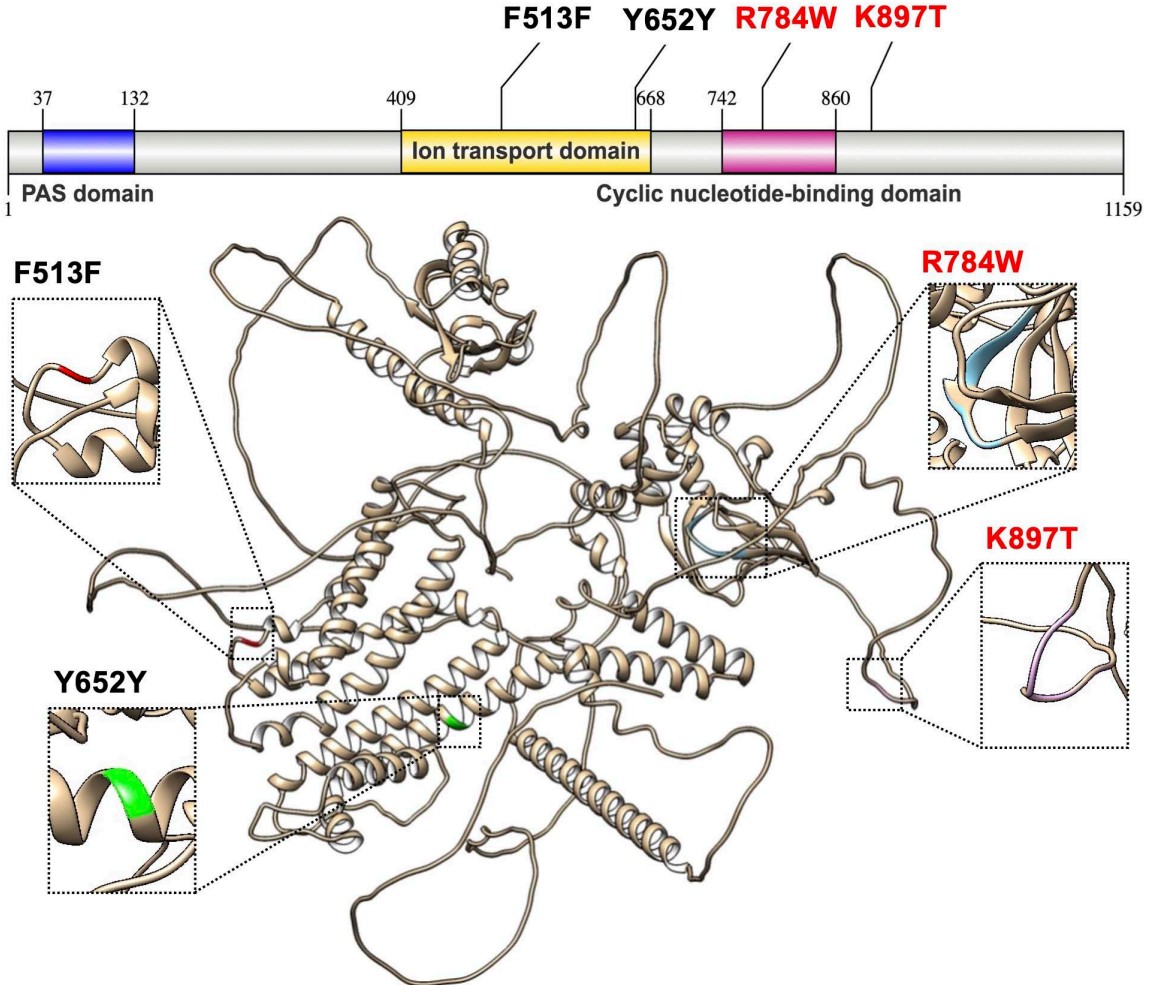

**Fig 3. The KCNH2 is 1159 aa long and contains three primary domains: PAS domain, ion transport domain, and cyclic nucleotide-binding domain.** The structure also highlighted the positions of four SNPs: 1539C>T (F513F), 1956T>C (Y652Y), 2350C>T (R784W), and 2690A>C (K897T).

**Table 5. Mutational effect prediction of two missense mutations 2350C>T (R784W) and 2690A>C (K897T) on the KCNH2 protein structure.**

| Gene | Codon and AA changes | Predicted stability change (ΔΔG) (kcal/mol) | | | | Location | Mutational effect |
|---|---|---|---|---|---|---|---|
| | | PremPS | mCSM | SDM | DUET | | |
| *KCNH2* | 2350C>T: R784W | **0.29** | **-0.435** | 0.73 | **-0.285** | Surface | Destabilizing |
| *KCNH2* | 2690A>C: K897T | **0.28** | **-0.188** | **-0.18** | 0.035 | Surface | Destabilizing |

stable, diminished half-life, and translated at a lower rate into hERG ion channel proteins compared to the native KCNH2 mRNA [55]. This is corroborated by Wang et al [56], who discovered a significant relationship that links the synonymous 1539C>T SNP to increased predisposition to atrial fibrillation in individuals of Han Chinese descent, which they attributed to left atrial enlargement. Hence, we postulate that the 1539C>T (F513F) synonymous substitution produces QTc lengthening effects via the above mechanisms, and thus, greater emphasis is required to further characterize the definitive link between synonymous SNPs and QTc interval prolongation.

**Table 6. The binding energy scores of methadone, levomethadone, and dextromethadone to the selected cavity/binding pocket containing SNPs as contact residues. F513F, Y652Y, and R784 are wild-type, while W784 is a mutant due to the substitution of Arg (R) to Trp (W) at position 784. ** indicates drug binding at the polymorphism site within contact residues.**

| KCNH2 | Binding pocket | Binding energy score (kcal/mol) | | |
|---|---|---|---|---|
| SNPs | Cavity | Methadone | Levomethadone | Dextromethadone |
| F513F (WT) | cavity_13_IVAPLYCDFQGTRESKHN ** | -6.9 | -6.9 | -6.8 |
| | cavity_50_VMLIADFYQPRTK ** | -6.9 | -6.7 | -6.5 |
| | cavity_52_ALPVCYQGEDRTSFKI ** | -6.9 | -6.6 | -6.5 |
| | cavity_75_DILFAPMRYVKTGS ** | -6.7 | -6.7 | -6.6 |
| Y652Y (WT) | cavity_5_SYRFLIVDCMANTGWH | -6.2 | -6.4 | -6.6 |
| | cavity_15_LFIYVMCKTAWPHS | -5.6 | -6.0 | -5.7 |
| | cavity_40_FLMVTAINYHSG ** | -5.8 | -6.1 | -6.1 |
| | cavity_76_FTYAVSLMIGCP | -5.8 | -5.9 | -5.5 |
| R784 (WT) | cavity_1_LIGSFRVYHPDATKNEMCQW | -6.3 | -6.0 | -6.1 |
| | cavity_12_NKHCSLDGFQTIRYAPMV | -6.1 | -6.0 | -6.0 |
| | cavity_20_KHARNSYCGQTDFILV | -6.0 | -6.0 | -6.0 |
| | cavity_68_ILDKNSGTFRPYHVA | -6.0 | -5.9 | -6.1 |
| | cavity_69_EKSIAFTNLRDHCG | -5.1 | -4.9 | -5.4 |
| | cavity_90_AIFTLDNKSGR ** | -3.7 | -3.8 | -3.7 |
| W784 (MT) | cavity_1_LIGSFRVYHPDATKNEMCQW | -6.0 | -5.9 | -5.9 |
| | cavity_12_NKHCSLDGFQTIRYAPMV | -5.4 | -5.8 | -5.8 |
| | cavity_20_KHARNSYCGQTDFILV | -6.1 | -6.2 | -6.0 |
| | cavity_68_ILDKNSGTFRPYHVA | -5.8 | -6.0 | -6.0 |
| | cavity_69_EKSIAFTNLRDHCG | -5.1 | -5.1 | -5.1 |
| | cavity_90_AIFTLDNKSGR ** | -4.3 | -4.3 | -4.8 |

Furthermore, there are 4 transcript variants for *KCNH2.* Using UCSC Genome Browse (https://genome.ucsc.edu/), all four SNPs are present in transcript variant 1 of *KCNH2,* which is its major transcript variant. Upon translation, this yields the isoform a of α subunit of $I_{Kr}$ potassium channel, the longest isoform of KCNH2 protein. In other transcript variants, however, not all four SNPs are present. For instance, rs1805123, which is located at exon 6 of *KCNH2*, is absent in transcript variant 2. Therefore, the precise functionality of non-major *KCNH2* splice variants should be examined to understand their effects on QTc prolongation.

Coupled with a molecular docking study, the 1539C>T (F513F) mutation produces best binding results by forming π-π stacking and three hydrophobic interactions with methadone, levomethadone and dextromethadone compared to other SNPs. Hydrophobic interactions are crucial for stabilizing drugs at binding sites [57] as they compensate for enthalpy-entropy balance, enabling hydrophilic residues to shed water molecules and facilitating hydrogen bonding and electrostatic attraction among them at the binding site [58]. Hydrophobic interactions result from water molecules occupying lipophilic binding sites, thus preventing hydrogen bonding with the receptor. Upon release from the hydrophobic pocket, the water molecules can form strong hydrogen bonds with bulk water [59], establishing that hydrophobic interactions of Phe513 drive *HCNH2* conformational changes upon methadone binding. Besides, π-π stacking facilitates diverse drug delivery and serves as a pivotal force in loading drugs into systems without altering their structural or functional properties [60]. The stacking is stabilized by the aromatic side chains present in the ligand-binding site due to the presence of phenylalanine, tryptophan, histidine, and tyrosine residues. Despite being considered a weak, non-covalent dispersion force, π–π interactions are critical in protein folding, thermal stability, and ligand binding [61]. Aromatic rings not only enhance

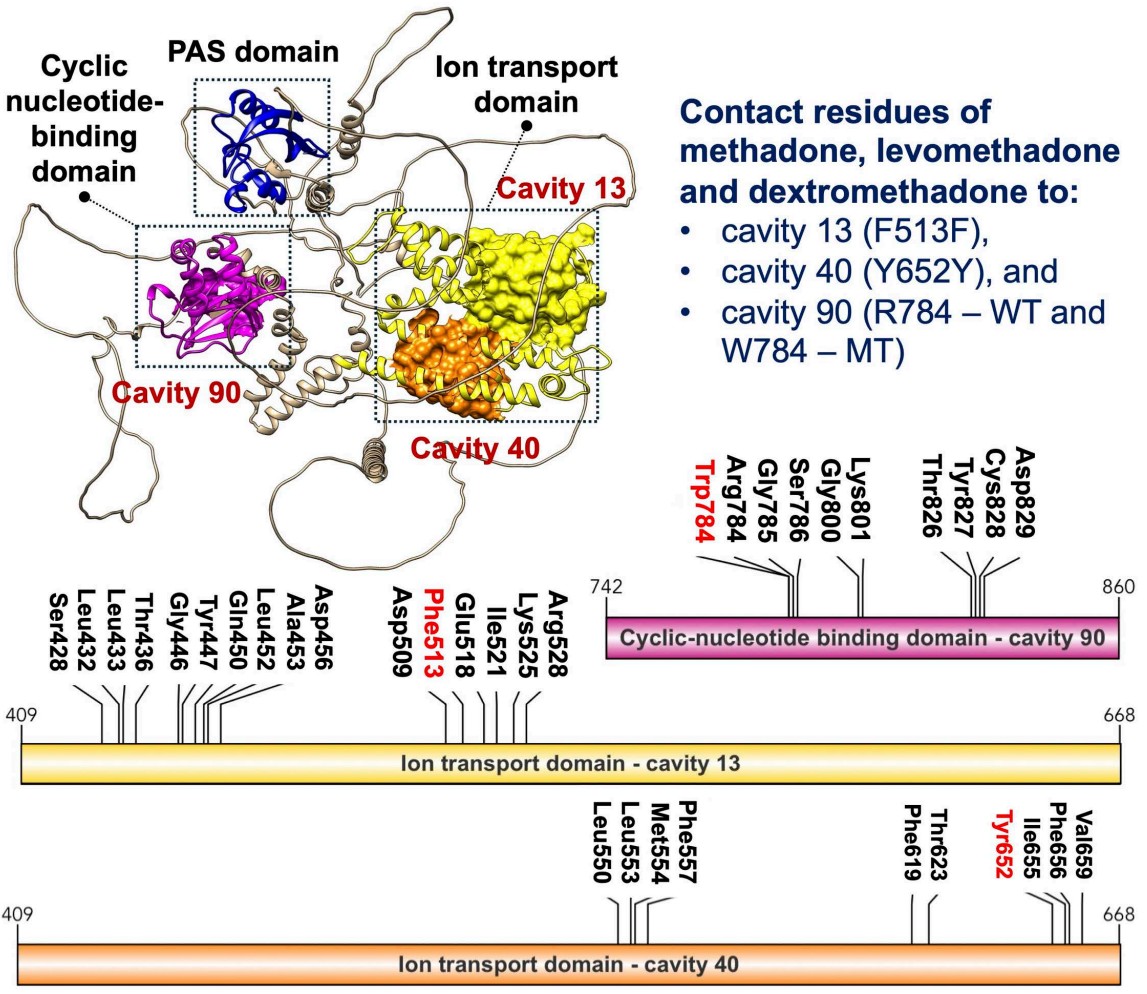

**Fig 4. Visualization of all contact residues of methadone, levomethadone, and dextromethadone within cavities 13 and 40 in the ion transport domain, and cavity 90 in the cyclic nucleotide-binding domain of the KCNH2 protein.**

drug design but also improve affinity and specificity, serving as the foundation for lead compound skeletons that can be refined to fulfill both target and off-target binding criteria [62]. Based on these arguments, our discoveries further supplement the findings by Titus-Lay and co-workers, who showed $QT_c$ prolongation was caused by both methadone isomers [63], instead of only S-isomers [64].

There are several study limitations that may bias the validity of our findings. Firstly, the gender effect on QTc could not be sufficiently adjusted in view of the restricted pool of female participants in our study cohort. This is highly inevitable due to most opioid-dependent users are male in Malaysia [65]. Next, we were unable to control the effects of population stratification and other genetic confounders (e.g., assortative mating, gene-environment interaction) due to our small study sample size. A study with a much larger sample size based on the genome-wide association study (GWAS) design is thus required to statistically adjust for the effects of genetic confounders adequately. Apart from those, the intra-observer reliability for QTc interval measurements was not assessed. Future studies should consider evaluating intra-observer variability using intraclass-correlation coefficient (ICC) to ensure QTc measurement consistency, especially when manual QTc measurements are performed by a single assessor. Besides, two of our studied SNPs violated Hardy-Weinberg

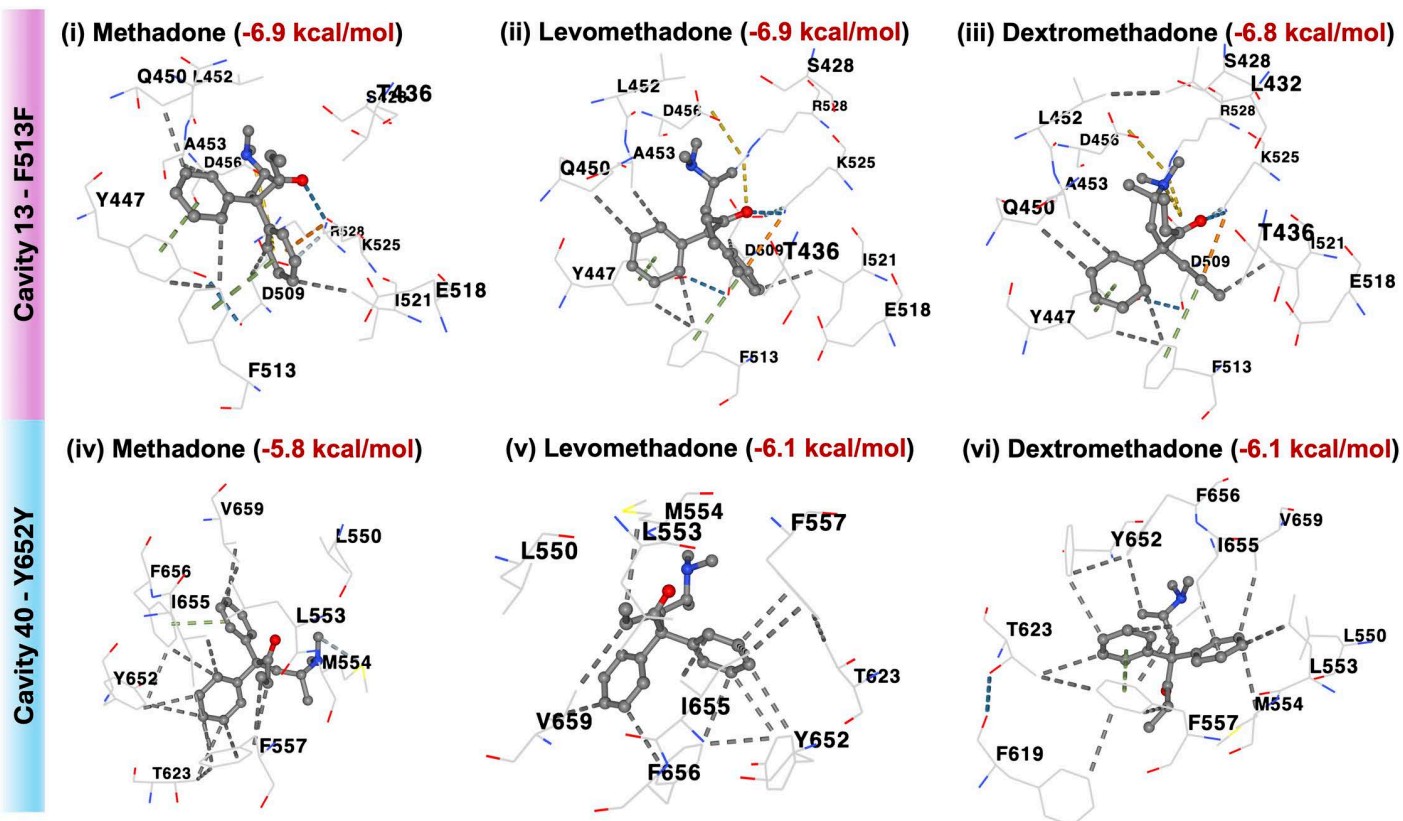

**Fig 5. Molecular docking of methadone, levomethadone, and dextromethadone on cavities 13 (F513F) and 40 (Y652Y) of KCNH2 using CB-DOCK2.**

equilibrium, and this calls our genotyping quality and our relatively small sample size into question. Furthermore, our small sample size and the cross-sectional design of this study may preclude the establishment of the causal relationship between *KCNH2* polymorphism and $QT_C$ prolongation due to lack of temporal order of the cause and effect. A prospective cohort design therefore may serve as a more reliable way of ascertaining the causality of such an association. Apart from that, the predictive model (equation 3) obtained is not properly validated. Ideally, a predictive model should be validated using a fresh population sample, but several internal validation techniques such as leave-one-out cross-validation (LOOCV), k-fold cross-validation, and bootstrapping are also good alternatives. These are now addressed in our ongoing research work. The generalizability of this study's findings is also restricted to the Kelantanese Malay ethnicity due to the exclusive racial composition of our study sample. However, our findings might be relevant for future studies conducted within similar ethnicity, which is also a major ethnicity in other South-East Asian (SEA) countries, for example, Indonesia, Brunei, Singapore, and Southern Thailand.

Our findings have several therapeutic ramifications. First, our discoveries offer a considerable impetus for recommending personalized treatment approaches for MMT patients who are homozygous carriers of the mutant-type genotype (1539TT) in managing their opioid dependence. The identified *KCNH2* polymorphisms may function as a critical predictor to tailor the treatment strategies of opioid dependence in MMT recipients to optimize the benefits and mitigate the adverse impacts associated with MMT. Therefore, in addition to regular electrolye and serum methadone trough level monitoring and correction for hypokalemia and hypomagnesaemia, clinicians should also screen for variations in the SNPs 1539C > T

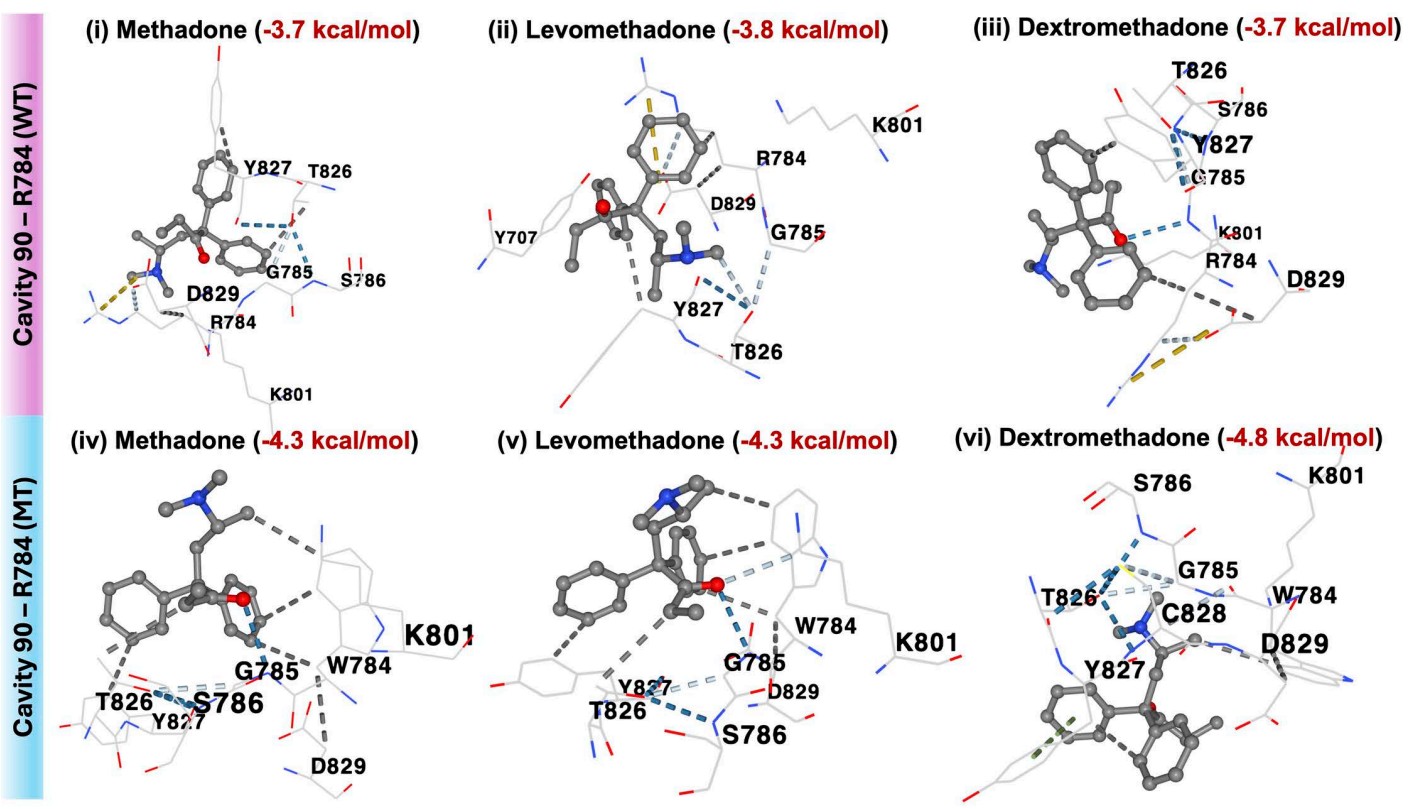

**Fig 6. Molecular docking of methadone, levomethadone, and dextromethadone to cavity 90 (R784 and W784) within the cyclic nucleotide-binding domain of WT and MT KCNH2 protein using CB-DOCK2.**

in the Kelantanese Malay MMT patients and those harbouring the TT genotypes should receive lower starting methadone dose and be closely monitored through routine ECG screening for QTc prolongation. Besides, our findings can also be incorporated with other existing prediction models [66] for predicting methadone-associated cardiotoxicity that can be specifically tailored for our Kelantanese Malay MMT patients, a population living in a state experiencing the highest number of drug abuse-related hotspots in Malaysia [67].

## Conclusion

In conclusion, this study identifies the synonymous 1539C>T polymorphism in *KCNH2* as a contributor to QTc prolongation in opioid-dependent MMT recipients. Molecular docking suggests that this variant may alter methadone binding within the hERG channel, potentially affecting cardiac repolarization. The developed predictive model, integrating genetic and clinical factors, may aid in risk stratification and individualized methadone dosing to prevent methadone-associated cardiotoxicity. These findings underscore the need for preemptive QTc monitoring and stringent pharmacogenetic screening to enhance MMT safety in high-risk populations.

## Supporting information

**S1 Table. Primers for the 1st and 2nd (Set A and Set B) PCR reactions.**
(DOCX)

**S1 Fig. The original gel electrophoresis image for Fig 2 in the main manuscript.**
(PDF)

**S2 Fig. Additional gel electrophoresis image for KCNH2 polymorphisms.**
(PDF)

## Acknowledgments

We posthumously thanked the late Professor Rusli Ismail and Dr. Muslih Abdulkarim Ibrahim for the initial conceptualization of this research.

## Author contributions

**Conceptualization:** Muhammad Irfan Abdul Jalal.

**Data curation:** Muhammad Irfan Abdul Jalal.

**Formal analysis:** Muhammad Irfan Abdul Jalal, Muhammad-Redha Abdullah-Zawawi, Nurfadhlina Musa, Basyirah Ghazali.

**Investigation:** Muhammad Irfan Abdul Jalal, Nurfadhlina Musa.

**Methodology:** Muhammad Irfan Abdul Jalal, Muhammad-Redha Abdullah-Zawawi, Basyirah Ghazali.

**Project administration:** Muhammad Irfan Abdul Jalal.

**Supervision:** Muhammad Irfan Abdul Jalal, Nasir Mohamad.

**Validation:** Nurfadhlina Musa.

**Visualization:** Muhammad Irfan Abdul Jalal, Muhammad-Redha Abdullah-Zawawi.

**Writing – original draft:** Muhammad Irfan Abdul Jalal, Muhammad-Redha Abdullah-Zawawi, Nurfadhlina Musa, Zalina Zahari, Nasir Mohamad.

**Writing – review & editing:** Muhammad Irfan Abdul Jalal, Muhammad-Redha Abdullah-Zawawi, Zalina Zahari, Nasir Mohamad.

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
