## [Decision Letter · Decision Letter 0]

24 Feb 2025

PONE-D-25-03646A synonymous KCNH2 polymorphism and methadone trough level influence QTc prolongation in Kelantanese Malay recipients of methadone maintenance therapy (MMT) in MalaysiaPLOS ONE

Dear Dr. Abdul Jalal,

Thank you for submitting your manuscript to PLOS ONE. After careful consideration, we feel that it has merit but does not fully meet PLOS ONE’s publication criteria as it currently stands. Therefore, we invite you to submit a revised version of the manuscript that addresses the points raised during the review process.

We look forward to receiving your revised manuscript.

Kind regards,

Nejat Mahdieh

Academic Editor

PLOS ONE

Journal Requirements:

“This research work was supported by the USM ‘Research University Grant (RUI)’ (Grant ID: 1001/PPSP/812056)”

4. Please include a caption for figure 1.

Reviewers' comments:

Reviewer's Responses to Questions

**Comments to the Author**

1. Is the manuscript technically sound, and do the data support the conclusions?

Reviewer #1: Yes

Reviewer #2: Yes

2. Has the statistical analysis been performed appropriately and rigorously? 

Reviewer #1: Yes

Reviewer #2: Yes

3. Have the authors made all data underlying the findings in their manuscript fully available?

Reviewer #1: Yes

Reviewer #2: Yes

4. Is the manuscript presented in an intelligible fashion and written in standard English?

Reviewer #1: Yes

Reviewer #2: Yes

5. Review Comments to the Author

Reviewer #1: The manuscript addresses an important clinical and pharmacogenomic topic, examining the association between KCNH2 polymorphisms and QTc intervals in opioid-dependent individuals undergoing methadone maintenance therapy (MMT). This study provides valuable insights into potential genetic factors influencing QTc prolongation, a well-recognized risk factor for methadone-associated cardiotoxicity. While the research is relevant and well-structured, certain areas require further clarification and refinement to enhance the manuscript’s quality and scientific rigor.

1. The manuscript does not provide sufficient demographic details about the study population, such as age, sex distribution, or comorbid conditions. Adding this information would help contextualize the findings and assess the generalizability of the results.

2. Were there any exclusion criteria beyond "free of cardiac structural defects"? This should be clarified.

3. It is unclear whether QTc intervals were corrected using Bazett’s formula, Fridericia’s formula, or another method. This should be explicitly stated, as the choice of correction formula could influence the results.

4. Was inter- or intra-observer variability assessed for QTc interval measurements? If not, this could be a limitation.

5. While the use of nested allele-specific PCR is mentioned, the specific conditions (e.g., primer sequences, cycling conditions, and quality control measures) are not detailed. Including this information would enhance reproducibility.

6. Were the genotypes in Hardy-Weinberg equilibrium (HWE)? This is a standard expectation for genetic association studies.

7. The molecular docking analysis, while intriguing, lacks sufficient detail. What software and algorithms were used? Were any controls or validation steps performed to ensure the accuracy of the docking results?

8. The docking results are not quantitatively compared across polymorphisms. Providing binding affinity scores or interaction energies for all four polymorphisms would allow for a clearer interpretation.

9. The discussion could better integrate the findings with previous research. Are there comparable studies in other populations or with other drugs that support or contradict these results?

10. The clinical implications of the findings (e.g., personalized methadone dosing or regular QTc monitoring) should be elaborated upon. How might these findings influence clinical practice?

11. The limitations section should explicitly acknowledge the relatively small sample size (n=111) and the cross-sectional design, which precludes causal inferences.

12. The study does not address potential population stratification or other genetic confounders, which could influence the observed associations.

13. Figures or visual representations of the molecular docking results would significantly enhance the manuscript’s clarity and impact.

14. The statistical results, such as confidence intervals and p-values, are presented in the text but would benefit from being summarized in a table for easier interpretation.

Reviewer #2: hank you to the authors for their manuscript. Below are some points for consideration:

he background information is relevant and well-supported by citations. However, there are several areas where improvements could enhance clarity and conciseness: 1. Length and Focus: The introduction is quite lengthy and could benefit from being more concise. Consider summarizing some points to maintain reader engagement and focus on the key objectives of the study. 2. Genetic Variability Discussion: The section discussing pharmacogenomics and genetic variations is informative but could be streamlined. Consider focusing on the most relevant genetic factors that relate directly to the study's objectives, rather than providing an extensive overview of multiple genes unless they are directly tied to your research question. 3. Repetitiveness: There are instances of redundancy, particularly in discussing QTc prolongation and its implications. Ensure that each point made adds new information or perspective to avoid repetition. Overall, while the introduction is thorough and informative, focusing on conciseness, clarity, and flow will significantly enhance its effectiveness in setting up the study's context and objectives.

The Methods section provides a detailed overview of the study design, participant selection, and sample size calculations.

The Results section presents a comprehensive analysis of the findings related to the QTc interval, KCNH2 protein structure, and the impact of specific SNPs on drug binding affinity. While the content is rich and informative, several areas could benefit from clarification and enhancement for improved readability and understanding:1:QTc Interval Findings: The initial statement regarding the mean QTc interval is clear; however, it would be helpful to provide context or reference values for what constitutes a clinically significant QTc prolongation. This would allow readers to better understand the implications of the 506 ms difference. 2. Protein Structure Models: The description of the protein structure models is detailed, but consider breaking up long sentences for better clarity.3.Conclusion of Findings: Conclude this section with a summary statement that ties together the key findings related to QTc prolongation, SNP effects on KCNH2 stability, and drug binding affinities. This will help reinforce the significance of your results and their implications for MMT recipients. Overall, this section effectively communicates important findings related to KCNH2 polymorphisms and their potential impact on QTc intervals in MMT recipients.

6. PLOS authors have the option to publish the peer review history of their article (what does this mean? ). If published, this will include your full peer review and any attached files.

**Do you want your identity to be public for this peer review?** For information about this choice, including consent withdrawal, please see our Privacy Policy .

Reviewer #1: No

Reviewer #2: **Yes: ** Mahdieh Soveizi

---

## [Author Response · Author response to Decision Letter 1]

11 Mar 2025

A) Comments from the editor

Editor

Response: We thank the editor for the comments. We have ensured that the revised manuscript conforms to the PLOS ONE's style requirements by removing the street address and ZIP codes from all authors' affiliations.

“This research work was supported by the USM ‘Research University Grant (RUI)’ (Grant ID: 1001/PPSP/812056)”

Please state what role the funders took in the study. If the funders had no role, please state: "The funders had no role in study design, data collection and analysis, decision to publish, or preparation of the manuscript.""

Response: We thank you the editor for comments. We would like to clarify that the funders do not have any other roles in the study apart from the provision of research funding. We have also declared and included the funder's role in the cover later as required by the editor (2nd paragraph; cover letter).

Response: We thank the editor for insightful comments. We have included the full ethics statement in the Methods section in lines 95-101, furnished with the following details:

a) The full name of the IRB / ethics committee: Universiti Sains Malaysia (USM)'s Human Research Ethics Committee (HREC), Kelantan, Malaysia.

b) Ethics approval number: USMKK/PPP/ Ethics.Com/2010(19)

c) Approval date: 01/02/2010.

Besides, we have also stated that written informed consent was obtained from every participant, prior to their inclusion into the study. Lines 95-101 now reads:

"The ethics approval for the research protocol was granted by the Universiti Sains Malaysia (USM)’s Human Research Ethics Committee (HREC), Kelantan, Malaysia (Reference number: USMKK/PPP/ Ethics.Com/2010(19), approval date: 01/02/2010). This study was conducted in accordance with the ethical principles outlined in the Declaration of Helsinki for research involving human participants and all participants provided written informed consent prior to their inclusion in the study."

4. Please include a caption for figure 1.

Response: We thank the editor for the comments. The caption for Figure 2 is actually for Figure 1. We have rectified the error accordingly and now the caption reads " Figure 1. The STROBE flow diagram summarizing the progress of each phase of this research" (Lines 109-111).

B) Comments from the reviewers

Reviewer 1

1. The manuscript does not provide sufficient demographic details about the study population, such as age, sex distribution, or comorbid conditions. Adding this information would help contextualize the findings and assess the generalizability of the results.

Response: We thank the reviewer for the comments. We have included the clinicodemographic characteristics of the participants in Table 1 (age, gender of the participants, SWOS scores, serum biochemical profiles, serum methadone trough, QTc intervals) (lines 333-335). We also mentioned that the participants did not have other comorbidities that may significantly affect the QTc intervals (lines 329-331).

2. Were there any exclusion criteria beyond "free of cardiac structural defects"? This should be clarified.

Response: We are grateful of the reviewer's pertinent comments. We have rephrased the abstract to include the full exclusion criteria of the study. The relevant part of abstract now reads "A cross-sectional study was conducted involving 111 patients with stable methadone dosage for at least 6 months attending several methadone clinics in Kelantan, Malaysia between March 2011 and October 2012. Those with cardiac structural defects, recipients of other QTc-prolonging pharmacotherapeutic agents, had aggressive behavior or other active psychiatric illnesses, chronic medical and surgical ailments and who were unable to communicate in Malay and English were excluded." (lines 28-34).

3. It is unclear whether QTc intervals were corrected using Bazett’s formula, Fridericia’s formula, or another method. This should be explicitly stated, as the choice of correction formula could influence the results.

Response: We thank the reviewer for such an insightful comment. We have clarified that the QTc intervals were calculated using the Fridericia's method in the abstract (line 33-34) and in the main text (lines 148-149).

4. Was inter- or intra-observer variability assessed for QTc interval measurements? If not, this could be a limitation

Response: We appreciate the reviewer’s comments regarding inter- and intra-observer variability in QTc interval measurements. As stated in the amended manuscript (lines 143-144), QTc intervals were measured by a single internist; therefore, inter-observer variability does not apply to this study. However, we acknowledge that intra-observer reliability was not assessed, and we have explicitly included this limitation in the discussion section and suggested possible remedies to overcome this limitation in future studies (lines 582- 585).

5. While the use of nested allele-specific PCR is mentioned, the specific conditions (e.g., primer sequences, cycling conditions, and quality control measures) are not detailed. Including this information would enhance reproducibility.

Response: Again, we thank the reviewer for such pertinent comments. The specific conditions. The details on the primer sequences, GC content, calculated melting temperature (Tm), size of PCR products (in base pairs), volume (µL) per reaction are provided in the supplementary Table S1 (to avoid the distortion of the manuscript flow and prevent the manuscript from becoming excessively long and unreadable). The design of the allele-specific PCR for KCNH2 detection and full PCR protocols are provided in the manuscript (lines 173-222). We believe the extent of the details that we provide ensures the reproducibility of our nested allele-specific PCR method.

6. Were the genotypes in Hardy-Weinberg equilibrium (HWE)? This is a standard expectation for genetic association studies.

Response: We thank the reviewer for such insightful remarks. To test the Hardy-Weinberg Equilibrium (HWE), we have used Levene and Haldane’s exact test based on the sum equally likely or more extreme samples (SELOME) method and Lindley’s α metric to evaluate the magnitude of departure from HWE. We chose these methods over the usual χ2 test for testing the HWE assumption since the standard χ2 relies on large sample (asymptotic) approximations, which become unreliable in small samples where expected genotype counts are low [1]. This can lead to inflated Type I error rates and inaccurate p-values, ultimately compromising the validity of the test. SELOME test improves upon this by employing a likelihood-based framework that accounts for small-sample bias and overdispersion, making it more reliable, particularly in cases with imbalanced genotype frequencies [2]. Lindley’s 𝛼 metric further refines the HWE assumption check by providing a continuous measure of deviation, offering a more informative alternative to dichotomous significance testing [3]. By addressing the inherent weaknesses of the chi-square test, both SELOME’s and Lindley’s methods provide more robust and precise evaluations of HWE, particularly in scenarios involving limited sample sizes or skewed genotype distributions.

We have provided further information on the testing of the HWE assumption in our revised manuscript (lines 338 - 346): "The genotype frequencies observed in this study were similar to those predicted by the HWE for 1539C>T (pexact(SELOME) = 0.4513) and 2690A>C (pexact(SELOME) = 0.1101) only, whilst the observed genotype frequencies for 1956T>C (minor allele frequency (MAF): 0.730) and 2350C>T (MAF: 0.036) significantly deviated from the expected genotype frequencies under HWE (pexact(SELOME) = <0.001 (Lindley’s α = 1.077) and 0.004 (Lindley’s α = 0.860), respectively. Full results are available upon request). Consequently, both 1956T>C and 2350C>T were excluded from further statistical analyses."

References

1. Waples RS. Testing for Hardy-Weinberg proportions: have we lost the plot? J Hered. 2015;106(1):1-19. doi: 10.1093/jhered/esu062.

2. Graffelman J. The number of markers in the HapMap project: some notes on chi-square and exact tests for Hardy-Weinberg equilibrium. Am J Hum Genet. 2010;86(5):813-8; author reply 818-9. doi: 10.1016/j.ajhg.2009.11.019.

3. Minelli C, Thompson JR, Abrams KR, Thakkinstian A, Attia J. How should we use information about HWE in the meta-analyses of genetic association studies? Int J Epidemiol. 2008;37(1):136-46. doi: 10.1093/ije/dym234.

7. The molecular docking analysis, while intriguing, lacks sufficient detail. What software and algorithms were used? Were any controls or validation steps performed to ensure the accuracy of the docking results?

Response: We appreciate the comment by the reviewer. In this study, we used CB-DOCK2 to perform docking analysis. We have also added the algorithm of CB-DOCK2 in facilitating the molecular docking study as recommended (Lines 303 – 305):

"CB-DOCK2 is a docking server that employs CurPocket, an algorithm based on protein surface curvature, to identify cavities and facilitate molecular docking using AutoDock Vina [43]."

Regarding the controls or validation steps, the binding energy of WT KCNH2 served as a reference, while the binding affinity of MT KCNH2 was compared against WT KCNH2 to determine the impact of polymorphisms on the binding stability of the drug and protein. We have revised the method for clarity (Lines 309 – 318):

"To assess the impact of KCNH2 polymorphisms on drug binding, the predicted structures of the MT and WT KCNH2 proteins were docked with methadone, dextromethadone, and levomethadone using CB-DOCK2, with contact residues for docking specified to the predicted active sites/cavities. All three drug structures were docked within the cavity using the same docking size and region to compare their binding affinities to the pocket of the MT and WT KCNH2 proteins. The binding energy (kcal/mol) of the MT and WT KCNH2 complexes with methadone, dextromethadone, and levomethadone was subsequently evaluated. The binding affinity of WT KCNH2 served as the control, while the binding affinity of MT KCNH2 was compared to that of WT KCNH2 to determine the impact of the polymorphism on drug-protein binding stability."

8. The docking results are not quantitatively compared across polymorphisms. Providing binding affinity scores or interaction energies for all four polymorphisms would allow for a clearer interpretation.

Response: Thank you for the comment by the reviewer. In response, we have provided the binding affinity of all three polymorphisms. K897T was excluded from molecular docking analysis as the mutation was not determined to be located at the binding pocket of KCNH2. We have presented the scores in Table 6 (lines 449 – 452), which include F513F and Y652Y as WT due to the unchanged amino acid. Additionally, R784 is referred to as WT due to the reference amino acid, while W784 is designated as MT KCNH2 due to the amino acid substitution from Arg (R) to Trp (W) at position 784. Our goal is to support the hypothesis that 1539T>C (F513F) is significantly associated with the mean QTc interval. Molecular docking results prove that KCNH2 F513F has better binding with methadone, levomethadone, and dextromethadone compared to other mutations. For clearer interpretation, we have added an explanation in the results section (Lines 454 – 461):

K897T was excluded from molecular docking analysis as the mutation was not determined to be located at the binding pocket of KCNH2. All three types of methadone exhibited more negative binding energy values to 1539C>T (F513F) compared to 1956T>C (Y652Y), 2350C>T (R784), and 2350C>T (W784). Overall, the drug binding energy score of F513F ranges from 6.5 to 6.9 kcal/mol, followed by Y652Y (5.8 – 6.1 kcal/mol), and R784 (3.7 – 3.8 kcal/mol). Out of the four mutations, W784 was also shown to have the highest binding energy score, suggesting a destabilizing effect on the KCNH2 protein, leading to ineffective drug binding (Table 5).

9. The discussion could better integrate the findings with previous research. Are there comparable studies in other populations or with other drugs that support or contradict these results?

Response: We appreciate the reviewer’s suggestion to better integrate our findings with previous research. In response, we have strengthened the discussion to provide clearer comparisons with prior studies.

The frequencies of KCNH2 polymorphisms (1539C>T and 2690A>C) observed in our study have been compared to those reported in previous studies, as detailed in Table 2 (lines 348 -350). Additionally, the effects of the 2690A>C polymorphism on QTc prolongation in MMT recipients align with findings by Hajj et al., showing a similar trend despite our results not reaching statistical significance due to the low frequency of CC genotype carriers (n=3 (2.7%); refer Table 1). This is further supported by studies by Bezzina et al. and Marjamaa et al., which demonstrated a link between the C allele and shorter QTc intervals (Lines 516–525).

As for the synonymous 1539C>T polymorphism, our study presents a novel finding, as no previous research has documented its association with QTc prolongation in MMT recipients in any other populations. We have emphasized this uniqueness in the revised discussion (lines 501 - 506).

We appreciate the reviewer’s feedback in helping us contextualize our findings within the existing literature.

10. The clinical implications of the findings (e.g., personalized methadone dosing or regular QTc monitoring) should be elaborated upon. How might these findings influence clinical practice?

Response: We appreciate the reviewer’s insightful suggestion regarding the elaboration of the clinical implications of our findings, particularly in relation to personalized methadone dosing and QTc monitoring. We have revised the Discussion section (lines 606-611). to highlight these clinical implications of our findings.

11. The limitations section should explicitly acknowledge the relatively small sample size (n=111) and the cross-sectional design, which precludes causal inferences.

Response: We thank the reviewer for such insightful remarks. We have added the reviewer's suggestions in the manuscript (lines 587- 590).

12. The study does not address potential population stratification or other genetic confounders, which could influence the observed associations.

Response: We thank the reviewer for such insightful comments. We have included the reviewer's suggestions as one of our study limitations and recommended that a larger population-based GWAS study should be carried out to address this limitation (lines 578 - 582).

13. Figures or visual representations of the molecular docking results would significantly enhance the manuscript’s clarity and impact.

Response: Thank you for the suggestion by the reviewer. In response, we have provided the figures for the molecular docking analysis of methadone, levomethadone, and dextromethadone (please refer to Figures 5 and 6). The presentation of the docking results was based on drug binding at the polymorphism site within contact residues and the lowest binding energy. Figure 5 demonstrates the docking results of methadon

---

## [Decision Letter · Decision Letter 1]

27 Mar 2025

A synonymous KCNH2 polymorphism and methadone trough level influence QTc prolongation in Kelantanese Malay recipients of methadone maintenance therapy (MMT) in Malaysia

PONE-D-25-03646R1

Dear Dr. Abdul Jalal,

We’re pleased to inform you that your manuscript has been judged scientifically suitable for publication and will be formally accepted for publication once it meets all outstanding technical requirements.

Kind regards,

Nejat Mahdieh

Academic Editor

PLOS ONE

Additional Editor Comments (optional):

Reviewers' comments:

Reviewer's Responses to Questions

**Comments to the Author**

1. If the authors have adequately addressed your comments raised in a previous round of review and you feel that this manuscript is now acceptable for publication, you may indicate that here to bypass the “Comments to the Author” section, enter your conflict of interest statement in the “Confidential to Editor” section, and submit your "Accept" recommendation.

Reviewer #3: All comments have been addressed

2. Is the manuscript technically sound, and do the data support the conclusions?

Reviewer #3: Yes

3. Has the statistical analysis been performed appropriately and rigorously? 

Reviewer #3: Yes

4. Have the authors made all data underlying the findings in their manuscript fully available?

Reviewer #3: Yes

5. Is the manuscript presented in an intelligible fashion and written in standard English?

Reviewer #3: Yes

6. Review Comments to the Author

Reviewer #3: This study found that the KCNH2 1539C>T polymorphism is associated with QTc prolongation in Malay opioid-dependent patients on methadone maintenance therapy (MMT), along with serum methadone, potassium, and magnesium levels. Molecular docking suggested strong binding between 1539C>T and methadone, highlighting the need for QTc monitoring to prevent methadone-induced cardiotoxicity in this population. In my opinion, reviewers' comments have beed addressed.

7. PLOS authors have the option to publish the peer review history of their article (what does this mean? ). If published, this will include your full peer review and any attached files.

**Do you want your identity to be public for this peer review?** For information about this choice, including consent withdrawal, please see our Privacy Policy .

Reviewer #3: No

---

## [Editor Report · Acceptance letter]

PONE-D-25-03646R1

PLOS ONE

Dear Dr. Abdul Jalal,

I'm pleased to inform you that your manuscript has been deemed suitable for publication in PLOS ONE. Congratulations! Your manuscript is now being handed over to our production team.

Kind regards,

on behalf of

Dr. Nejat Mahdieh

Academic Editor

PLOS ONE